# High Glucose Induces in HK2 Kidney Cells an IFN–Dependent ZIKV Antiviral Status Fueled by Viperin

**DOI:** 10.3390/biomedicines10071577

**Published:** 2022-07-01

**Authors:** Alawiya Reslan, Juliano G. Haddad, Philippe Desprès, Jean-Loup Bascands, Gilles Gadea

**Affiliations:** 1Unité Mixte Processus Infectieux en Milieu Insulaire Tropical, Plateforme Technologique CYROI, Université de la Réunion, INSERM U1187, CNRS UMR 9192, IRD UMR 249, 94791 Sainte Clotilde, La Réunion, France; alawiyareslan@hotmail.com (A.R.); juliano.haddad@univ-reunion.fr (J.G.H.); philippe.despres@univ-reunion.fr (P.D.); 2Unité Mixte Diabète Athérothrombose Thérapies Réunion Océan Indien, Plateforme Technologique CYROI, Université de la Réunion, INSERM U1188, 94791 Sainte Clotilde, La Réunion, France; 3Institut de Recherche en Cancérologie de Montpellier, Université de Montpellier, INSERM U1194, IRCM, F-34298 Montpellier, France

**Keywords:** flavivirus, high glucose, interferon, ISG, kidney cells, viperin, Zika virus

## Abstract

Zika virus (ZIKV) is an emerging mosquito-borne flavivirus that rapidly became a major medical concern worldwide. We have recently reported that a high glucose level decreases the rate of Zika virus (ZIKV) replication with an impact on human kidney HK-2 cell survival. However, the mechanisms by which cells cultured in a high glucose medium inhibit ZIKV growth remain unclear. Viperin belongs to interferon-stimulated genes (ISG) and its expression is highly up-regulated upon viral infection, leading to antiviral activity against a variety of viruses, including flaviviruses. As such, viperin has been shown to be a major actor involved in the innate immune response against Zika virus (ZIKV). Our present study aims to further characterize the involvement of viperin in ZIKV growth inhibition under high glucose concentration (HK-2^HGC^). We show for the first time that endogenous viperin is over-expressed in HK-2 cells cultured under high glucose concentration (HK-2^HGC^), which is associated with ZIKV growth inhibition. Viperin knockdown in HK-2^HGC^ rescues ZIKV growth. In addition, our results emphasize that up-regulated viperin in HK-2^HGC^ leads to ZIKV growth inhibition through the stimulation of IFN-β production. In summary, our work provides new insights into the ZIKV growth inhibition mechanism observed in HK-2 cells cultured in a high glucose environment.

## 1. Introduction

Viruses are widely distributed in nature. They can be found freely distributed in almost all ecosystems in a cell-dependent manner. The cell represents the host in which viruses replicate. When cells are infected with a virus, they activate several effective defense pathways that form the basis of the innate immunity. Infected cells produce interferons (IFNs), powerful antiviral molecules that protect neighboring cells from infection and thus limit virus spread (reviewed in [1]). All mammalian nucleated cells carry this system, which, when activated, results in the induction of several interferon-stimulated genes (ISGs), some of which have direct antiviral properties. Among ISGs, viperin (virus inhibitory protein endoplasmic reticulum associated interferon inducible) or Rsad2 (radical SAM domain-containing 2), identified almost 20 years ago, has been described as an inhibitor of viral replication by different mechanisms [2]. The activation of viperin is highly dependent on functional IFN signaling pathways [3,4], and independently in a direct manner through IRF3 (interferon regulatory factor 3), activator protein 1 (AP-1), and IRF1 (interferon regulatory factor 1) [2,5,6].

Infection with many human viruses induces the activation of viperin, which is able to limit viral infection in most of the cases, with the first description of viperin overexpression in human cytomegalovirus (HCMV) infected cells [7,8]. Subsequently, viperin has been shown to be involved in other viral infection restriction, such as influenza virus, hepatitis C virus (HCV), Sindbis virus (SINV), human immunodeficiency virus (HIV), West Nile virus (WNV), Japanese encephalitis virus (JEV) [6,9,10,11,12,13], and more recently, dengue virus (DENV) and Zika virus (ZIKV) [13,14,15,16,17].

The antiviral activity of viperin appears to be diverse and polyvalent. In the case of ZIKV infection, viperin can catalyze the conversion of cytidine triphosphate (CTP) into 3′-deoxy-3′,4′-didehydro-CTP (ddhCTP), which serves as a chain terminator during ZIKV replication [18]. Viperin can also interact with the viral non-structural protein NS3, thereby promoting its degradation, which could also explain the restriction of ZIKV replication [16]. In addition, the interaction of viperin with Golgi-dependent soluble protein trafficking induces the release of immature capsids [19]. Furthermore, beyond its contribution to viral restriction and innate immunity, viperin exerts complex and diverse cellular functions [20,21,22].

Our recent study aimed to improve our knowledge on ZIKV replication in human HK-2 kidney cells under high glucose concentration (HK-2^HGC^) compared to the HK-2 cells under normal glucose condition (HK-2^NGC^) [23]. We reported, for the first time, that an elevated glucose level can inhibit ZIKV infection in HK-2 cells by directly acting on the replication step without affecting early infection stages. In this study, we try to decipher the molecular mechanisms responsible for ZIKV growth inhibition under high glucose conditions. To this end, taking into account that HK-2^HGC^ overexpress the transcription factors, IRF1 and IRF3, known to stimulate several IFN-stimulated genes including viperin [24,25], we ask whether viperin could participate in the mechanism of ZIKV growth inhibition. Our data show, for the first time, that the antiviral mechanism against ZIKV infection is driven by an overexpression of viperin, which in turn stimulates IFN-β secretion.

## 2. Materials and Methods

### 2.1. Cells, Virus, Plasmids, and Reagents

Human kidney proximal tubular HK-2 cells (CRL-2190, ATCC, Manassas, VA, USA) were grown in Dulbecco’s modified Eagle’s medium (DMEM) supplemented with D-glucose (5.6 mM), 10% heat inactivated fetal bovine serum (FBS), 2 mmol.L^−1^ L-Glutamine, 100 U/mL penicillin, and 0.1 mg.mL^−1^ streptomycin and 0.5 µg.mL^−1^ of fungizone (Amphotericin B) (PAN Biotech, Aidenbach, Germany) at 37 °C under 5% CO_2_. In prior experiments, cells were adapted to high glucose (25 mM) for a 10-day period to allow metabolic reprogramming. Glucose concentrations were chosen based on our recent study [23] and on previous published works [26,27,28,29]. The number of cells seeded in the two conditions was adjusted based on the cell growth kinetics study to avoid the multiplicity of infection (MOI) bias during infections (as previously described in [23]). A549 cells (Invivogen Inc, Toulouse, France) were grown in DMEM medium supplemented with 10% heat-inactivated FBS, non-essential amino acids, 10 µg.mL^−1^ blasticidin and 100 mg.mL^−1^ zeocin (InvivoGen, Toulouse, France). HEK-Blue IFN-α/β cells were grown in DMEM supplemented with 10% FBS, Blasticidin, Normocin, and Zeocin. Antibiotics and SEAP substrate (QUANTI-Blue^TM^) were purchased from InvivoGen (Toulouse, France).

Mouse anti-viperin antibody clone MaP.VIP was purchased from Sigma-Aldrich (Saint Quentin Fallavier, France) and the mouse anti-DDDDK tag mAb (anti-FLAG_tag_ antibody) from Abcam (Cambridge, UK). Zika virus (ZIKV) was detected by using mouse anti-E monoclonal antibody 4G2 (RD Biotech (Besancon, France)). Donkey anti-mouse Alexa Fluor 488 secondary antibody was purchased from Invitrogen (Carlsbad, CA, USA), and goat anti-mouse and goat anti-rabbit immunoglobulin-horseradish peroxidase (HRP) conjugated secondary antibodies from Abcam (Cambridge, UK). DAPI was purchased from Euromedex (Souffelweyersheim, France). The rabbit anti-α tubulin polyclonal Ab was purchased from Santa Cruz Biotechnology (Dallas, TX, USA). SiRNA-VP (TAGAGTCGCTTTCAAGATA) was designed as previously described [30]. SiRNA-VP and scrambled siRNA (TTCTCCGAACGTGTCACGT) were purchased from Genecust, France.

The recombinant plasmids (rVP^wt^ and rVP^mut^) were verified by sequencing (Genecust, France). Recombinant ZIKV expressing the GFP reporter gene (ZIKV^GFP^) was previously described [31]. Cells were routinely infected with ZIKV at the MOI of 1 plaque-forming units (PFU) per cell.

### 2.2. RT-qPCR

Quantification of viperin RNA was performed by RT-qPCR. For this, total cellular RNA of HK-2 cells was extracted from cells with RNeasy kit (Qiagen, Hilden, Germany) according to the manufacturer’s instructions and reverse transcribed using random hexamers pd(N)6 and M-MLV reverse transcriptase (Life Technologies, Carlsbad, CA, USA) at 42 ◦C for 50 min. cDNA were amplified using 0.2 µM of each primer (forward primer: GATGTTGGTGTAGAAGAAGC and reverse primer: CCAATCCAGCTTCAGATCAG) and 2X Absolute Blue qPCR SYBR Green Low ROX Master Mix (ThermoFisher, Waltham, MA, USA) on a CFX96 Real-Time PCR Detection System (Bio-Rad, Life Science, Hercules, CA, USA). The CFX96 program was used to calculate the threshold cycle (Ct) for each well in the exponential phase of amplification. The results obtained by RT-qPCR of our interest gene were normalized to the housekeeping gene GAPDH in each condition.

### 2.3. Transfection Experiment

The recombinant plasmids (rVP^wt^ and rVP^mut^) and siRNA-VP were designed as previously described [17,30], respectively, and verified by sequencing (Genecust, Luxembourg). Cells grown in 24-well plates were transfected with recombinant plasmids or siRNA-VP for a 24 h period. Transfections were performed with Lipofectamine 3000 and Lipofectamine RNAiMAX (Invitrogen, Carlsbad, CA, USA) according to the manufacturer’s instructions.

### 2.4. Immunofluorescence Assay

HK-2 cells were fixed (PFA, 3.7%) at 24 h post-infection or 24 h post-transfection and permeabilized for 5 min (PBS 1× 0.15% Triton X-100). Cells were then stained for viperin using anti-VP (1:1000 in PBS-BSA 2%) for 1 h at room temperature. Mouse monoclonal antibody anti-flag was used to detect wild-type rVP^wt^ and mutant rVP^mut^. The goat anti-mouse Alexa Fluor 488 IgG was used as secondary antibody (1/1000) for 30 min in the dark. The cell nuclei were delineated using DAPI staining. Coverslips were mounted in Vectashield and fluorescence was observed using a Nikon Eclipse E2000-U and Nikon Eclipse TI2-S-HU (confocal microscopy). Images were captured using a Hamamatsu ORCA-ER camera and NIS-Element AR (Nikon, Champigny-sur-Marne, France) imaging software (Nikon, Champigny-sur-Marne, France). The results were quantified by ImageJ software.

### 2.5. Flow Cytometry Analysis

HK-2 cells were seeded in 6-well plates and infected with ZIKV^GFP^ at a multiplicity of infection (MOI) of 1. The cells were then harvested at 24 h post infection or transfection, fixed with 3.7% PFA for 10 min, permeabilized with 0.15% Triton X-100 in PBS for 5 min, and then blocked with PBS-BSA 1% for 10 min. For the detection of rVPs and viperin endogen, cells were labeled with anti-FLAGtag antibody (dilution 1:2000) and anti-VP antibody (dilution 1:1000), respectively, for 1 h at room temperature. Goat anti-mouse Alexa Fluor 488 IgG was used as the secondary antibody (1/1000) for 30 min. Cells were then subjected to a flow cytometric analysis using a CytoFLEX flow cytometer (Beckman Coulter, Brea, CA, USA). Results were analyzed using CytExpert v2.1 software (Beckman Coulter Villepinte, France.

### 2.6. MTT Assay

MTT (3-[4-dimethylthiazol-2-yl]-2,5-diphenyltetrahzolium bromide) colorimetric assay was used to evaluate cellular metabolic activity. HK2 cells grown in 96-well plates were transfected with rVP^wt^ and rVP^mut^. At 48 h post-transfection, culture supernatants were discarded and 20 µL of 5 mg.mL^−1^ MTT solution (Sigma-Aldrich, France) were added to the cells. After 2 h at 37 °C, the MTT solution was discarded, and formazan crystals were solubilized in 50 µL of dimethyl sulfoxide (DMSO). Absorbance was measured at 570 nm with background subtraction at 690 nm.

### 2.7. LDH Assay

The cytopathic effect of rVP^wt^ and rVP^mut^ was determined by Lactate Dehydrogenase (LDH) colorimetric assay measuring LDH release. Supernatants of rVP^wt^ and rVP^mut^ transfected cells were collected at 48 h post-transfection. LDH assay was performed using CytoTox 96 nonradioactive cytotoxicity assay (Promega, Madison, WI, USA), according to the manufacturer’s instructions. Absorbance was measured at 490 nm with background subtraction at 690 nm.

### 2.8. Western Blot Assay

Cells were lysed in RIPA buffer. Cell lysates were then precleared by centrifugation at 12,000× *g* for 10 min at 4 °C. Protein samples were heated or not for 5 min at 95 °C in Laemmli sample buffer. Proteins in cell lysates were separated by SDS-PAGE, then transferred onto nitrocellulose membranes (Hybond-ECL; Amersham, Frieburg im Breslgau, Germany) that were incubated with blocking buffer (5% skim milk in 1 × PBS -TWEEN) after Ponceau S (Ponceau Red) staining. Immunoblots were probed for viperin (mouse anti-viperin antibody, 1/1000) and ZIKV-E protein (mouse anti-E monoclonal antibodies 4G2, 1/1000). Following incubation with primary antibodies, the membranes were incubated with the corresponding peroxidase conjugated anti-mouse antibodies. Immunoblots were revealed with ECL detection reagents (Amersham), as recommended by the manufacturer.

### 2.9. Quantification of Type I IFN by SEAP Activity

HEK-blue IFN-α/β cells were used for the detection of type I interferons (IFN-α and IFN-β) by activating of ISGF3 pathway, inducing SEAP secretion in the supernatant. HEK-Blue cells were seeded in 96-well plates at a concentration of 20,000 cells/well. A total of 100 µL of HK-2 cell supernatants were added to HEK-blue cells and incubated at 37 °C and 5% CO_2_ for 24 h. On the next day, 20 µL of supernatant was transferred to a 96-well plate and 180 µL of QUANTI-Blue^TM^ solution were added and incubated up to 3 h at 37 °C. Optical density (OD) at 655 nm was measured using a microplate reader (Omega, Ortenberg Germany).

### 2.10. ELISA Assay

The 96-well plates (MaxiSorp, Thermo Fisher Scientific Inc., USA) were coated with the supernatants of HK-2 cells at 4 °C overnight. After 3 washes with PBS-Tween (PBST), plates were incubated with 5% skim milk at 37 °C for 1 h, and then incubated with primary antibody rabbit anti-IFNβ for 2 h. After 5 washes with PBST, immunoblots were incubated with avidin-HRP for 1 h. The reactions were developed using 3, 3′, 5, 5′-tetramethylbenzidine (TMB). A total of 2 M H_2_SO_4_ was used to stop the reaction. OD values of each well were measured at 450 nm.

### 2.11. Statistical Analysis

Statistically significant differences between conditions were analyzed with a one-way ANOVA test, two-way ANOVA test, and unpaired *t*-test. All values were expressed as mean  ±  SD of at least three independent experiments performed in triplicate. All statistical analysis were conducted using Graph-Pad Prism software, version 8.0 (GraphPad Software, San Diego, CA, USA). Degrees of significance are indicated on the figures as follows: * *p*  <  0.05; ** *p*  <  0.01; *** *p*  <  0.001, ns  =  not significant.

## 3. Results

### 3.1. HK-2^HGC^ Cells Display a High VP Protein Expression Level

We have recently reported an inhibition of ZIKV infection in HK-2^HGC^ cells, which was associated with less viral genome copies [23]. Because viperin is an antiviral protein whose expression is highly up-regulated during viral infections via IFN-dependent and/or IFN-independent pathways, we first investigated whether viperin was expressed in HK-2-cells and whether viperin expression was modified in a high glucose medium. As shown in Figure 1A, viperin mRNA is weakly expressed in HK-2 grown in normal-glucose concentration (HK-2^NGC^), but significantly overexpressed in HK-2^HGC^ cells.

As viperin protein expression is cell type dependent [9,18,32], we first checked whether viperin protein was expressed in HK-2 cells. To this end, we quantified its level by flow cytometry showing the mean fluorescence intensity (MFI) of positive cells using an anti-VP antibody (Figure 1B). Viperin protein is overexpressed in HK-2^HGC^ cells compared to HK-2^NGC^ cells. These data were visually confirmed by immunofluorescence (Figure 1C) and quantified by ImageJ (% of positive HK-2^NGC^ or HK-2^HGC^ cells are 15.6 ± 4% and 39.3 ± 5%, respectively) (Figure 1D). Thus, HK-2^HGC^ cells display a high viperin protein expression level, suggesting that endogenous viperin overexpression in these conditions could be involved in the anti-ZIKV activity shown in our recent study [23].

### 3.2. Overexpression of Viperin in HK-2^NGC^ Cells Inhibits ZIKV Infection

In order to assess the ability of viperin to inhibit ZIKV infection of HK-2 cells, we used a vector plasmid pcDNA3 expressing wild-type rVP^wt^ tagged with a C-terminal FLAG epitope. To rule out any experimental artifacts, we used a negative control mutant rVP^mut^ (as previously described in [17]). We first analyzed the expression of the two rVP plasmids in HK-2^NGC^ cells at 24 h after transfection. Representative images of the transfected cells stained with the anti-FLAG_tag_ antibody (Figure 2A) as well as analysis by flow cytometry analysis (Figure 2B) show similar expression levels for rVP^wt^ (30.3 ± 2.4%) and rVP^mut^ (27.8 ± 3.2%). At 48 h post-transfection, we observed no effect for rVP^wt^ or a moderate, but significant reduction in cell metabolic activity and cell membrane integrity for rVP^mut^ as measured by MTT and LDH assays, respectively (Appendix A).

Then, we investigated whether the expression of a recombinant viperin in HK-2 cells results in ZIKV infection inhibition under normal glucose conditions. HK-2 cells were transfected with rVP^wt^ or rVP^mut^ for a 24 h period followed by infection with ZIKV^GFP^ carrying the reporter gene of the green fluorescent protein (GFP) at MOI = 1 for another 24 h. The analysis of the GFP protein expression in HK-2 cells by flow cytometry showed that rVP^wt^ transient expression significantly reduced the percentage, around 65%, of ZIKV infected cells (Figure 2C). As expected, no effect was observed with the vehicle or the rVP^mut^. These data strongly suggest the involvement of viperin in the inhibition mechanism of ZIKV in high glucose conditions.

### 3.3. Viperin Knockdown in HK-2^HGC^ Cells Rescues ZIKV Replication

Our previous work [23] demonstrated that an elevated glucose level decreases ZIKV replication in HK-2 cells. The above data suggest an interplay between viperin overexpression and ZIKV growth inhibition in HK-2^HGC^ cells. To test this hypothesis and because viperin expression is hardly detectable in HK-2^NGC^ cells, viperin was knocked down in HK-2^HGC^ cells by transfection with siRNA-VP (si-Viperin) [30]. A scrambled siRNA (si-Control) was used as negative control. HK-2 ^HGC^ cells were transfected with siRNA for 24 h followed by infection with ZIKV^GFP^ for another 24 h. Cells were then lyzed and immunoblots targeting ZIKV-E and viperin proteins were realized using anti-E pan flavivirus and anti-viperin antibodies, respectively, and anti-alpha tubulin antibodies as control. As shown in Figure 3A–C and by normalizing the results to tubulin expression, si-Viperin decreases the expression of viperin in HK-2 cells and this was associated with a threefold increase in ZIKV E protein levels compared to control knockdown, which was not observed in NGC condition where the level of ZIKV E protein did not change before and after transfection with si-VP (data not shown).

Taken together, these results on viperin overexpression in HK-2^NGC^ and knockdown in HK-2^HGC^ cells demonstrate that viperin is a key element of the ZIKV infection inhibition mechanism.

### 3.4. Paracrine Antiviral Effect of Viperin in HK-2^HGC^ Cells

Surprisingly, whereas the transfection levels of rVP^wt^ plasmid in HK-2 cells are around 30% (Figure 2A,B), we could observe a 65% reduction in ZIKV infection (Figure 2C), suggesting that molecules acting at the paracrine level can be found in supernatants of HK-2 cells overexpressing viperin to control ZIKV infection.

To test this hypothesis, we transferred supernatants of rVP^wt^ or rVP^mut^ transfected HK-2 cells on naive HK-2^NGC^ cells at different time points pre- and post-infection and monitored ZIKV^GFP^ infection by flow cytometry 24 h post-infection (Figure 4A). Interestingly, we observed a significant inhibition (54%) of ZIKV infection in HK-2 cells when treated with supernatants of cells transfected with rVP^wt^ plasmid at 6 h pre-infection (Figure 4B) compared to cells treated with supernatants of non-transfected (vehicle) or rVP^mut^ transfected cells. The reduction in ZIKV infection was still significant (52%) up to 2 h pre-infection (Figure 4C). In contrast, when added 2 h post-infection, supernatants from rVP^wt^ transfected HK-2 cells had only a moderate and not significant antiviral effect (Figure 4D), and this ZIKV inhibiting effect was completely lost 6 h after infection (Figure 4E).

Next, we investigated whether molecules implicated in ZIKV growth inhibition were also present in the supernatants of HK-2^HGC^ cells. For this purpose, we transferred supernatants of HK-2^HGC^ cells collected at 24 h, 48 h, or 72 h period, onto naive HK-2^NGC^ cells at different time points (2 h, 6 h, and 24 h) before ZIKV^GFP^ infection. Cells were then subjected to a flow cytometry analysis to evaluate the percentage of GFP-expressing cells at 24 h post-infection. The results show a significant ZIKV growth inhibition after treatment up to 2 h pre-infection with the supernatant of HK-2^HGC^ cells collected at 72 h period compared to controls, i.e., supernatants from HK-2^NGC^ cells or naive cultured medium (Figure 4F). Interestingly, the supernatants of HK-2^HGC^ cells collected at 24 or 48 h were able to significantly control ZIKV infection up to 6 h before ZIKV-GFP infection (Appendix A).

Furthermore, we wondered whether this antiviral effect, acting through the supernatants of transfected cells, was specific to HK-2 cells or not. To this end, another cell line, the human lung carcinoma A549 cell line, was used to test the inhibitory effect of supernatants at 2 h pre- and post-infection. In pre-infection experiments, we observed an inhibition of ZIKV infection, while less strong than in HK-2 cells, and we totally lost the inhibitory effect in post-infection experiments (Appendix A). Our data indicate that both supernatants from HK-2 cells overexpressing viperin or grown in high glucose conditions lead to ZIKV growth inhibition in two different cell lines.

### 3.5. Viperin Has a Paracrine Function through IFN-β in HK-2 Cells under High Glucose Conditions

Thus, we hypothesize that molecules secreted following viperin overexpression enhance cellular antiviral activities through autocrine and paracrine effects. A previous study by Kumar et al. [33] showed that ZIKV was able to block type I and type III IFN signaling pathways. In their study, authors also found that the addition of IFNs before ZIKV infection blocked ZIKV replication. Conversely, ZIKV replication was unaffected by addition of IFNs after the initiation of a productive infection. Accordingly, we thought that viperin could also exert its antiviral effect in HK-2^HGC^ cells through IFN secretion. To check this out, we indirectly detected IFNs in supernatants of HK-2^HGC^ cells as well as in viperin (rVP^wt^ or rVP^mut^ plasmid) transfected cells. Importantly, it has been shown that IFN-β was the first and major type I interferons expressed and produced following virus infection [34], so we investigated the presence of IFN-β in the supernatants. To this end, supernatants were transferred for 24 h on HEK-blue IFN-α/β cells, which are used for the detection of type I interferons (IFN-α and IFN-β). The results showed that viperin overexpression leads to a high IFN level (Figure 5A). Importantly, we found that supernatants of HK-2^HGC^ cells contain even highest rates of IFNs than in viperin transfected cells. To further confirm this observation, we performed an ELISA assay to measure IFN-β concentration in supernatants (Figure 5B). Interestingly, we obtained similar results to those observed with HEK blue IFN-α/β cells, i.e., higher levels of IFN-β in supernatants of HK-2^HGC^ cells than in viperin-overexpressing cells (Figure 5B).

Finally, we went on to prove that IFNs enhance viperin antiviral activity in the extracellular environment. For this, we treated the supernatants of rVP^wt^ or rVP^mut^ transfected cells or non-transfected HK-2^NGC^ and HK-2^HGC^ cells with an anti-IFN-β antibody. We then removed the supernatants, which were transferred on naive HK-2^NGC^ cells 2 h before ZIKV^GFP^ infection for a 24 h period. Finally, cells were subjected to a flow cytometry analysis to quantify the percentages of infected cells. We found that the blockage of IFN-β with a specific antibody leads to the loss of the paracrine effect of viperin and rescues ZIKV infection in HK-2^HGC^ cells and in rVP^wt^ transfected cells compared to their respective control conditions (Figure 5C). Taken together, our data demonstrate that viperin-mediated IFN-β secretion under high glucose conditions drives ZIKV growth inhibition in HK-2 cells.

## 4. Discussion

The host-innate response to viral infection is essential to limit viral replication and promote the inflammatory processes through the expression of interferon-stimulated genes (ISGs). Viperin is one of the few ISGs that has been described as a restriction factor for a variety of viruses, which include multiple members of the flaviviridae family. This protein plays an emerging role in modulating innate immune signaling and inhibits viral replication in a variety of cell types by multiple mechanisms. Thus, the induction of viperin expression in ZIKV-infected cells with an antiviral activity against ZIKV replication was reported by others [15,16,17]. In this study, we demonstrated that high glucose induces the overexpression of endogenous viperin in human renal proximal tubule epithelial (HK-2) cells, which turns out to inhibit ZIKV infection. This inhibition was demonstrated to act at the ZIKV replication step in our previous work [23]. This is, to our knowledge, the first study describing in kidney cells an interplay between high glucose and viperin expression to inhibit ZIKV infection. Viperin expression is normally stimulated by type I (α and β), II (γ), and III (λ) IFNs and infection by various viruses. IFN-α and -β are potent inducers of viperin in a majority of cell types. A recent study has shown that viperin can directly be induced independently of IFN pathways, by interferon regulatory factor IRF1 and 3 [35,36]. Another study showed that these interferon regulatory factors are up-regulated in HK-2^HGC^ cells human kidney cells [24,25]. These data corroborate our results, showing that viperin is up-regulated in HK-2 cells cultured in the same experimental conditions. Since some cell lines, such as Hela or HEK293T cells, do not naturally express viperin [9,18], we investigated the expression of natural or artificial viperin protein in our HK-2 cell based model. The recombinant plasmid of viperin, rVP^wt^ (previously described in [17]), transfected in HK-2^NGC^ cells showed that viperin exerts a potent antiviral activity against ZIKV. We also found that the expression of a viperin mutant, rVP^mut^, resulted in a marked increase in cellular toxicity, whereas rVP^wt^ shows no cytotoxicity effect in HK-2 cells. The reduction in metabolic activity and loss of membrane integrity observed with rVP^mut^ may be due to a defect in the essential cellular functions of viperin. These results helped us to characterize the effect of the endogenous viperin over-expression in HK-2^HGC^ cells. Thus, we confirmed the role of endogenous viperin to inhibit ZIKV infection in HK-2^HGC^ cells by knocking down its expression. The si-RNA mediated knockdown of viperin increases ZIKV infection in HK-2 cells to a level comparable to the one obtained under low glucose conditions.

In addition to its direct antiviral activity on viral proteins, recent data suggest that viperin has an impact on the innate antiviral signaling. In our current study, we noticed that the rate of ZIKV infection inhibition obtained by the overexpression of viperin, after transfection with rVP^wt^ plasmid, is considerable compared to the rate of HK-2 cells expressing this exogenous viperin, suggesting the intervention of other factors capable of amplifying the antiviral activity. We demonstrated that the up-regulation of viperin induces the production of paracrine acting molecules in the supernatant. These molecules can then modulate innate immune signaling and inhibit ZIKV replication before the virus initiates its productive cycle.

Several studies have demonstrated that viperin itself can stimulate IFN-β induction by interacting with the signal mediators IRAK1 and TRAF6 and by phosphorylating the transcription factor IRF7 [37,38]. This then stimulates the transcription of IFN-β that will be secreted and signal both autocrine and paracrine by binding to its receptor. It has recently been shown that viperin knockdown decreases IRF7 recruitment to the nucleus, causing a down-regulation of IFN-β gene expression [39]. Interestingly, we could demonstrate from our overexpression experiments (HEK blue and ELISA) that recombinant VP, in HK-2^NGC^ cells, was able to induce interferon production and that this IFN production takes a major role in the observed ZIKV infection inhibition.

In this context, we confirmed the indirect anti-ZIKV function of viperin, by showing that IFN-β has an important role associated with the up-regulation of viperin in HK-2^HGC^ cells to limit ZIKV infection, suggesting that IFN-β was stimulated by endogenous viperin to reduce ZIKV replication. However, in HK-2^HGC^ cells, our data do not allow to clearly establish which comes first, viperin or IFN-β. Indeed, we are facing a feedback loop since, on the one hand, it has been shown that HGC can stimulate both IFN-β and viperin expression and, on the other hand, it is also known that viperin can stimulate IFN-β and vice versa. Although we demonstrated that si-Viperin reduced the inhibitory effect on ZIKV replication, we cannot rule out the possibility of the intervention of IFN-β. So, further studies blocking interferon are needed to clarify this point.

Surprisingly, although it was not associated with ZIKV inhibition, we observed a moderate IFN-β production in vehicle and rVP^mut^ transfected cells, which has to be related to the lipofectamine effect reported by Guo et al. [40]. Thus, further works are needed to better understand the induction pathways in these conditions.

Viperin has been reported to inhibit a wild variety of DNA and RNA viruses, including flaviviruses (hepatitis C virus (HCV), Zika virus (ZIKV), West Nile virus (WNV), and dengue virus (DENV)) [10,13,15,16,17]. In vitro studies have shown that kidney cells, such as human glomerular podocytes, renal glomerular endothelial cells, and mesangial cells, are highly permissive to ZIKV infection and replication, serving as reservoirs for virus dissemination [41]. In Zika virus patients, no clinical manifestations of renal disease, functional or structural damage, to the kidney have been described to date, although there is a persistent long-term viral shedding in the urine [42]. This may confirm the presence of activated factors in the kidney, particularly under conditions of hyperglycemia that limit viral infection.

Although this work is based on in vitro experiments, it could be placed in a more general context, namely, diabetic pathology. Flavivirus, such as Dengue, West Nile, Yellow fever, and Zika, have a kidney tropism [43] and a link with kidney diseases have been reported for dengue virus (DENV) and West Nile virus (WNV) [44,45,46,47]. Although it is known that a pathological environment such as hyperglycemia may lead to a depressed acquired immune response [48], it is surprising to note that most of the studies looking at the role of comorbidities in the outcome of flavivirus infection are only related to DENV and WNV infections. To the best of our knowledge, no relevant data for YFV and ZIKV are yet available. Regarding DENV and WNV infections, data are still controversial. Indeed, some studies have reported that comorbid diseases (diabetes, hypertension, and chronic kidney disease) lead to a more severe outcome to dengue and WNV infection [49,50,51,52], whereas other studies found no relationships [50,53,54]. Taken together, it appears that more clinical studies should be undertaken to better understand the impact of comorbidities on flavivirus (Dengue, West Nile, Yellow fever, and Zika) infection consequences.

## 5. Conclusions

In conclusion, we demonstrated that high glucose conditions up-regulate viperin in kidney cells, leading to a decrease in ZIKV growth, via a viperin mediated IFN-β pathway. As other flavivirus have been associated with renal disease in diabetic context, it will be interesting to evaluate the implication of viperin in the mechanisms of flavivirus-associated renal disease progression.

## Figures and Tables

**Figure 1 biomedicines-10-01577-f001:**
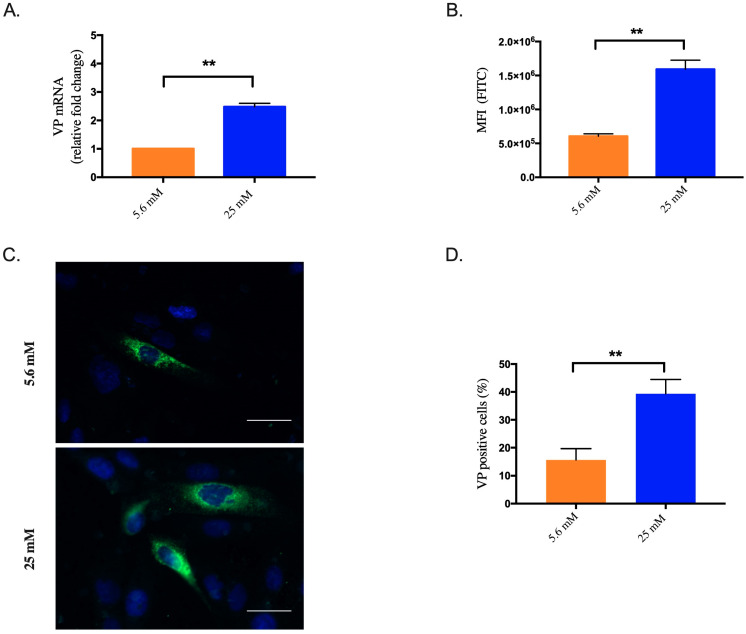
High glucose concentration induces endogenous viperin overexpression in HK-2 cells. (**A**) Viperin (VP) mRNA expression in HK-2^NGC^ cells (5.6mM) or HK-2^HGC^ cells (25mM) was detected using quantitative polymerase chain reaction. (**B**) Detection of endogenous viperin protein by flow cytometry. (**C**) Viperin protein detected by immunofluorescence staining in HK-2 cells. (**D**) Quantitative analysis of VP immunofluorescence by ImageJ software. Viperin (green) and nuclei (blue) were visualized by confocal fluorescence microscopy. Results are shown as means ± SD of three independent experiments. Unpaired t-test shows that differences are statistically significant (** *p* < 0.01). Scale bar represents 40 µm.

**Figure 2 biomedicines-10-01577-f002:**
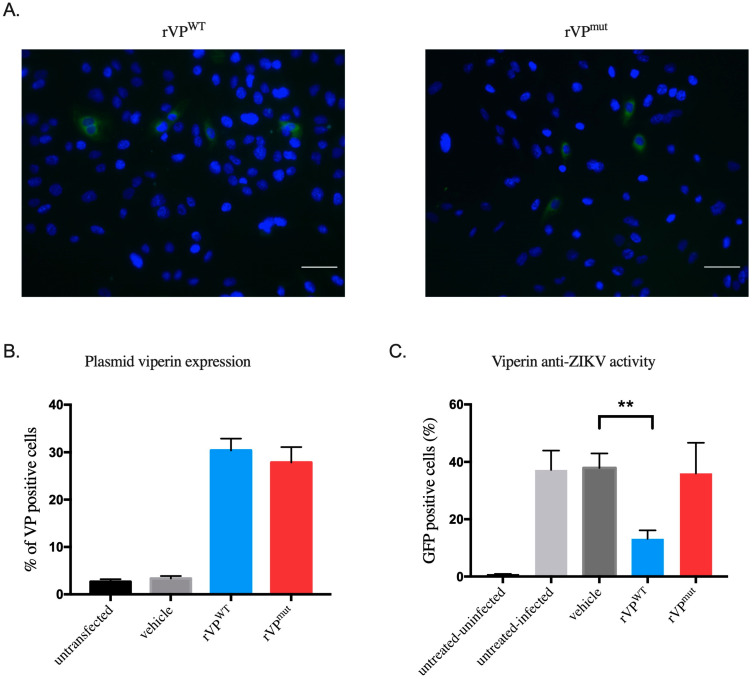
Anti-Zika virus activity of recombinant viperin in HK-2^NGC^ cells. Expression of recombinant wild-type viperin (rVP^wt^) and recombinant mutant viperin (rVP^mut^) proteins in HK-2^NGC^ cells detected by immunofluorescence assay (**A**) and flow cytometry (**B**). Viperin (green) and nuclei (blue) were visualized by fluorescence microscopy. (**C**) Anti-ZIKV activity of rVP^WT^ in HK-2^NGC^ cells. HK-2 cells were infected with ZIKV^GFP^ and GFP-positive cells were quantified by flow cytometry at 24 h post-infection. Results are shown as means ± SD. Unpaired t-test shows that differences are statistically significant (** *p* < 0.01). Scale bars represent 100 µm.

**Figure 3 biomedicines-10-01577-f003:**
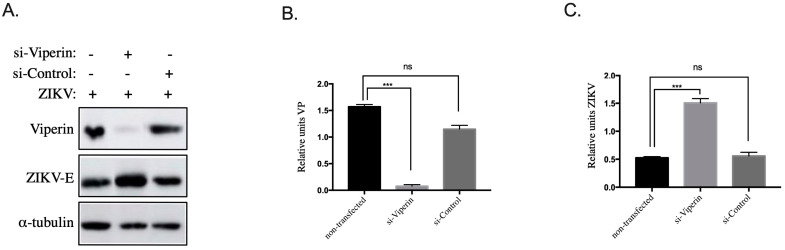
Overexpression of endogenous viperin in HK-2^HGC^ cells suppresses ZIKV infection. (**A**) HK-2^HGC^ cells were transfected with siRNA-VP (si-Viperin) or scrambled siRNA (si-Control) for 24 h and then infected with ZIKV^GFP^ at a MOI of 1 for another 24 h. Immunoblots show viperin (VP) and ZIKV-E proteins, as well as α-tubulin as loading control. Viperin protein and ZIKV-E quantification in (**B**,**C**), respectively, were performed by normalizing to tubulin from the results obtained in (**A**). Data are representative of at least three independent experiments. Results are shown as means ± SD. Unpaired t-test shows that differences are either non-significant (ns) or statistically significant (*** *p* < 0.001).

**Figure 4 biomedicines-10-01577-f004:**
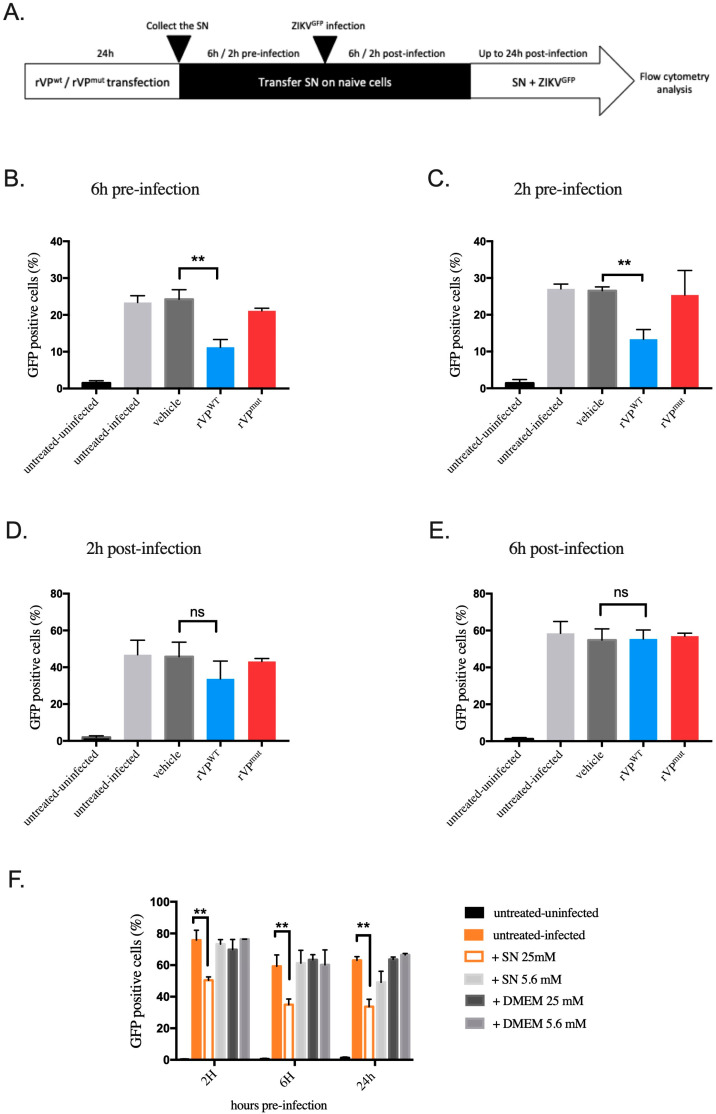
Paracrine inhibitory effect of supernatants from HK-2 cells overexpressing recombinant viperin (rVP) on ZIKV infection. (**A**) Experimental design to characterize the antiviral activity of cell supernatant (SN) from HK-2 transfected with rVP (**B**–**E**). The supernatants of HK-2 cells transfected with rVP^wt^ or rVP^mut^ were transferred onto naive HK-2^NGC^ cells at (**B**) 6 h pre-infection, (**C**) 2 h pre-infection, (**D**) 2 h post-infection, and (**E**) 6 h post-infection. (**F**) Supernatants of HK-2^HGC^ cells or a 72 h period were transferred on naive HK-2^NGC^ cells at 2 h, 6 h, and 24 h pre-infection. For all these conditions, cells were then infected with ZIKV^GFP^ for 24 h followed by flow cytometry analysis. SN 5.6 mM: supernatant of HK-2^NGC^ cells; DMEM 25 mM: high glucose medium; DMEM 5.6 mM: low glucose medium. Data are representative of at least three independent experiments. Results are shown as means ± SD. Unpaired t-test and two-way ANOVA show that differences are either non-significant (ns) or statistically significant (** *p* < 0.01).

**Figure 5 biomedicines-10-01577-f005:**
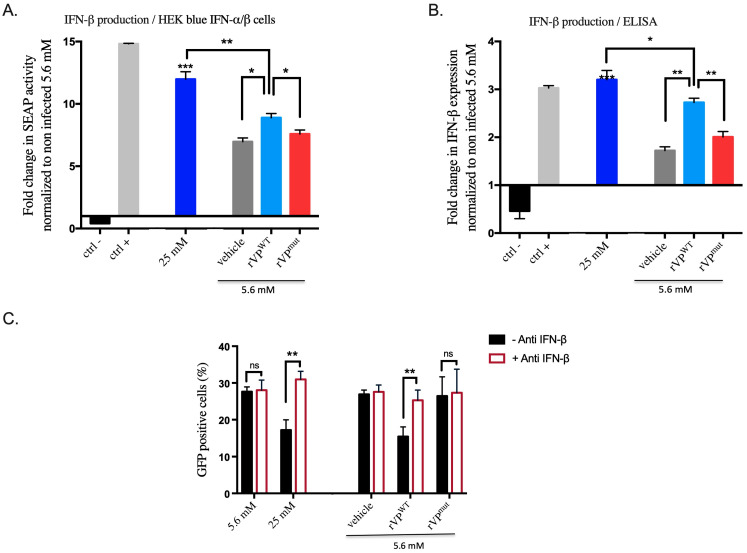
Viperin overexpression mediated IFN-β secretion in HK-2^HGC^ cells exerts an anti-ZIKV activity. (**A**) HK-2^NGC^ and HK-2^HGC^ cells media were transfected with rVP^wt^ or rVP^mut^, or non-transfected as control. To evaluate type I interferons by using SEAP activity, supernatants were transferred onto HEK-blue IFN-α/β cells for a 24 h period with DMEM (ctrl−) and IFN-β (ctrl+) as negative and positive controls, respectively. (**B**) Quantification of IFN-β in the supernatant of transfected or HK-2^NGC^ and HK-2^HGC^ cells was performed by the ELISA assay. In (**A**,**B**), results were represented as means ± SD. (**C**) Supernatant of transfected HK-2 cells were transferred on naive HK-2^NGC^ cells in the presence or not of anti-IFN-β for 2 h followed by ZIKV^GFP^ infection for a 24 h period. Cells were subjected to flow cytometry analysis. Data are representative of at least three independent experiments. Results are shown as means ± SD. One-way ANOVA and two-way ANOVA show that differences either non-significant (ns) or statistically significant (* *p* < 0.05, ** *p* < 0.01 and *** *p* < 0.001).

## Data Availability

Not applicable.

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
