# Peer review of "High Glucose Induces in HK2 Kidney Cells an IFN–Dependent ZIKV Antiviral Status Fueled by Viperin"

_biomedicines, 2022, doi:10.3390/biomedicines10071577_

Round 1

Reviewer 1 Report

In the following manuscript by Reslan and colleagues, the authors described how high levels of glucose affect the replication of Zika virus (ZIKV) in human kidney cells. This effect seems to be mediated by the increased expression of viperin in the presence of high glucose which leads to the stimulation of IFN-β production and the inhibition of ZIKV infection. This effect could be transferred to naïve HK cells by using conditioned supernatant obtained from HK cells grown under high glucose culture conditions or VPWT transfected, which also results in inhibition of ZIKV infection until two hours before viral exposure.

Viperin is a well-known interferon-inducible antiviral protein, responsible for the establishment of an innate antiviral response to a variety of viral infections. In fact, its contribution to modulating flavivirus infections as well as other groups of viruses have been previously reported. Additionally, its role in regulating the cell metabolism including the induced expression of the glucose transporter GLUT4, which results in increased glucose import and translocation to the nucleus of the glucose-regulated transcription factor ChREBP, has been also described during viral infection.

Although viperin has been shown to inhibit many viruses, the molecular antiviral mechanism is not clear and appears to differ between viruses. Here, following a series of experiments in vitro, the authors could propose a potential effect of high glucose conditions in inhibiting the infection with ZIKV after modulating the expression of viperin that results in an IFN-type I antiviral state. Despite the interesting findings, there are several gaps in the methodology that must be clarified to better support the results as well as the conclusions suggested by the authors as following described:

1.       In Figure 1A, when evaluating the effect of glucose level on the mRNA expression of viperin, how was the viperin mRNA level determined? How was concentration normalized? Did authors consider detecting the housekeeping mRNA to compare to? Can the authors please clarify how was this assay performed and how were RNA copy numbers determined? This is not explained either in Methods or Figure legend. Please explain.

2.       Considering that apparently no normalization step was performed in the mRNA quantification of Viperin, could these differences in mRNA expression obtained at 5.6- and 25-mM of glucose be explained by the growing rate of the HK cells on the culture system used? Please clarify this point.

3.       In Figure 1C, can authors look for a better representative image? The numbers of viperin expressing cells (green) vs the number of cells per visual area do not tight up with the VP positive cells (%) shown in Figure 1D. Besides, the confluency of the cell monolayers on the two images looks quite different between 5.6- and 25-mM of glucose.

4.       Additionally, can the authors explain how the percentage of positive cells was calculated, based on the surface area analyzed, the number of counted cells, etc. This is not specified either in Methods or the figure legend.

5.        Again, can authors improve the representative images included in Figure 2A showing the expression of both VP proteins, WT and Mut? Also, how was the efficiency of transfection and the percentage of VP expressing cells determined in these assays? These are important details at the time of performing and interpreting the results obtained after the transfection assay.

6.       Also, was transfection performed under conditions of high (25mM) or low (5mM) glucose concentration? These are important details not included in methods or figure legends.

7.       Regarding the ZIKV inhibition assays on HK-VPWT & VPMut transfected cells, how reliable is it to test viral replication using a transfection GFP reporter system on already transfected cells? Can authors corroborate these results using an infectious ZIKV strain?

8.       Also, can others specify what is depicted in each section of the Figures? There are many details missing, not explained, in the Figures, for instance, the number of events collected in Figure 2C, what mix stands for, etc.

9.       Following question number 6, were viral inhibition assays depicted in Figure 2 performed under conditions of high (25mM) or low (5mM) glucose concentration after transfection?

10.   Did the authors evaluate an additional viral marker than E to demonstrate viral infection and/or replication, for instance, NS3 or NS2A proteins? Infection was measured 24 hours post-infection which opens the possibility of detecting only the virus that was attached to the cell membrane and not a virus that replicated during that time. In fact, NS3 proteins of distinct flavivirus including ZIKV and TBEV have been shown to interact with viperin, and to be significantly degraded by viperin after viral infection, which correlated well with the antiviral activity of viperin. Please discuss it and check for NS3 or any other non-structural protein.

11.   Together with what the authors showed in Suppl. Figures 2 and 3 (paracrine effect of viperin) which step of the viral replication cycle do authors suggest is viperin inhibiting ZIKV infection? viral attachment, entry, replication, exit?

12.   Also, were Relative units for VP (Figure 2E) and ZIKV (Figure 2F) determined based on densitometry analyses of alpha-tubulin? This is not explained in either the Methods or figure legend.

13.    Additionally, it is confusing that some experiments were performed using HKHGC and others using HKNGC, the latter not previously introduced in the text, till line 253. Can authors better explain why they did this?

14.   The last part of the results section is interesting as the authors explored the effect of viperin on different human cells such as A549, a human lung carcinoma cell line; however, the antiviral effect on A549 cells was only explored by conditioned supernatants obtained from transfected cells and not directly on the A549 cells. It would be interesting to explore whether this viperin effect is specific to distinct cell lines from distinct tissue origins such as kidney (HK cells) and lung (A549 cells).  

Author Response

We thank the reviewer for valuable suggestions.

  1. In Figure 1A, when evaluating the effect of glucose level on the mRNA expression of viperin, how was the viperin mRNA level determined? How was concentration normalized? Did authors consider detecting the housekeeping mRNA to compare to? Can the authors please clarify how was this assay performed and how were RNA copy numbers determined? This is not explained either in Methods or Figure legend. Please explain.

We changed the Y axis to « VP mRNA (relative fold change) because the amount of VP mRNA quantified in each condition has been normalized to the amount of housekeeping gene mRNA (GAPDH). A sentence clarifying this issue was added in the method section (in the new version of the manuscript) (line 117)

  1. Considering that apparently no normalization step was performed in the mRNA quantification of Viperin, could these differences in mRNA expression obtained at 5.6- and 25-mM of glucose be explained by the growing rate of the HK cells on the culture system used? Please clarify this point.

The number of HK-2 cells seeded under these 2 conditions was adjusted. We took in consideration the cells growth kinetics of HK-2 cells under each condition (as previously described by Reslan et al., 2020 / ref23). A sentence explaining this matter was added in the methods section (in the new version of the manuscript (line 82). In addition, and to be sure that we do not quantify more VP mRNA in 25 mM, we normalized the amount of VP mRNA detected to the amount of housekeeping gene mRNA (GAPDH) in each condition.

  1. In Figure 1C, can authors look for a better representative image? The numbers of viperin expressing cells (green) vs the number of cells per visual area do not tight up with the VP positive cells (%) shown in Figure 1D. Besides, the confluency of the cell monolayers on the two images looks quite different between 5.6- and 25-mM of glucose.

We understand the concern of the reviewer. The images in figure 1C are representative of 10 images taken in each condition. In this figure, we aimed to demonstrate that endogenous VP is detected under our conditions, which is quite hard as VP protein is unstable in the cytoplasm (not an artifact). Consequently, we zoomed onto the cells to provide a better visualization of the staining, without quantitative goal. The percentage of VP positive cells represented in figure 1D is the result of the counting of VP positive cells in all 10 images.

  1. Additionally, can the authors explain how the percentage of positive cells was calculated, based on the surface area analyzed, the number of counted cells, etc. This is not specified either in Methods or the figure legend.

We agree with reviewer 1 on this matter. We determined the percentage of positive cells using ImageJ software. The number of positive cells was divided by the total number of cells. A sentence clarifying this issue was added in the figure legend (line 229) and in the method section (line 135).

  1. Again, can authors improve the representative images included in Figure 2A showing the expression of both VP proteins, WT and Mut? Also, how was the efficiency of transfection and the percentage of VP expressing cells determined in these assays? These are important details at the time of performing and interpreting the results obtained after the transfection assay.

We do apologize if this was unclear. The transfection efficiency was demonstrated in figure 2B with the percentage of the VP positive cells quantified by flow cytometry analysis of transfected cells under our 2 conditions. This was mentioned in figure 2B legend (line 275) and in the result section 3.2 (line 241).

  1. Also, was transfection performed under conditions of high (25mM) or low (5mM) glucose concentration? These are important details not included in methods or figure legends.

To clarify this point, we separated figure 2 into 2 new figures (fig. 2 with transfected HK-2NGC and fig.3 with HK-2HGC).

  1. Regarding the ZIKV inhibition assays on HK-VPWT& VPMut transfected cells, how reliable is it to test viral replication using a transfection GFP reporter system on already transfected cells? Can authors corroborate these results using an infectious ZIKV strain?

In this article, we used an infectious virus expressing the GFP reporter gene (not a GFP reporter system) inserted in juxtaposition with the capsid protein. Thus the level of intracellular GFP represents well the level of other viral proteins expressed in the cell (Gadea et al., 2016 / ref 31)

  1. Also, can others specify what is depicted in each section of the Figures? There are many details missing, not explained, in the Figures, for instance, the number of events collected in Figure 2C, what mix stands for, etc.

In cytometry experiments, we usually analyzed 10000 events. The reviewer has propably mistaken the word ‘mock” by “mix”. For more clarity, we changed “mock” to “vehicle” in all figures.

  1. Following question number 6, were viral inhibition assays depicted in Figure 2 performed under conditions of high (25mM) or low (5mM) glucose concentration after transfection?

As this was subjected to misunderstanding, we separated figure 2 into 2 new figures (fig. 2 with transfected HK-2NGC and fig.3 with HK-2HGC).

  1. Did the authors evaluate an additional viral marker than E to demonstrate viral infection and/or replication, for instance, NS3 or NS2A proteins? Infection was measured 24 hours post-infection which opens the possibility of detecting only the virus that was attached to the cell membrane and not a virus that replicated during that time. In fact, NS3 proteins of distinct flavivirus including ZIKV and TBEV have been shown to interact with viperin, and to be significantly degraded by viperin after viral infection, which correlated well with the antiviral activity of viperin. Please discuss it and check for NS3 or any other non-structural protein.

Again, we understand the concern of the reviewer. However, the amount of GFP protein was measured 24 hours post infection so the GFP level, showed in our results, is a direct reporter of viral replication after a complete viral cycle in the cells (24h post-infection). In addition, if the virus fails to infect the cells, the amount of E protein from the inoculum is not detectable by western blot (100000 PFU used in the assay). Lastly, the attached non-internalized virus does not remain stable for 24 hours until the cell lysis.

  1. Together with what the authors showed in Suppl. Figures 2 and 3 (paracrine effect of viperin) which step of the viral replication cycle do authors suggest is viperin inhibiting ZIKV infection? viral attachment, entry, replication, exit?

In our previous study, we showed that glucose acts on the replication step of ZIKV infection (Reslan et al.,2020 / ref 23). This was mentioned in the introduction section (line 66). In addition, it has been shown in the literature that VP blocks the virus at the replication level (different levels - see introduction) and IFN is also known in the literature to block viral replication.

  1. Also, were Relative units for VP (Figure 2E) and ZIKV (Figure 2F) determined based on densitometry analyses of alpha-tubulin? This is not explained in either the Methods or figure legend.

We agree with the reviewer on this matter. Thus, a sentence explaining this point was added in the legend of figure 3 and the result section (line 293).

  1. Additionally, it is confusing that some experiments were performed using HKHGCand others using HKNGC, the latter not previously introduced in the text, till line 253. Can authors better explain why they did this?

A sentence was added in the introduction section (line 63) to clearly define these two abbreviations

  1. The last part of the results section is interesting as the authors explored the effect of viperin on different human cells such as A549, a human lung carcinoma cell line; however, the antiviral effect on A549 cells was only explored by conditioned supernatants obtained from transfected cells and not directly on the A549 cells. It would be interesting to explore whether this viperin effect is specific to distinct cell lines from distinct tissue origins such as kidney (HK cells) and lung (A549 cells).  

In our present study we aimed to determine the role of VP protein in kidney cells (HK-2). This study is a follow-up of our previous study in which we determined the impact of HGC on ZIKV infection in kidney cells (HK-2) as a described reservoir of ZIKV and as a model to study renal diabetic disorders. In addition, we previously described the antiviral effect of VP against ZIKV in A549 cell line (Vanwalscappel et al., 2019 / ref 17)

Reviewer 2 Report

In this follow-up study, the authors describe an explanation for their observation that HK2 kidney cells grown in high glucose environment are able to suppress ZIKV replication by inducing viperin expression, and that this is related to IFN production.

This manuscript is written well and clear, the methods and results sections are described adequately and performed accurately. The conclusions are well-supported by the observations. There are some improvements possible related to experimental analysis or WB conditions and in the discussion section.

General remarks

-        - The structure of the manuscript is somewhat unclear. Could the section numbers and figures be homogenized? For example, Fig. 2 could be split in two separate figures according to section 3.2 and 3.3 (Fig.2 A-C and Fig. D-F), since this involves either NGC or HGC cells. Also, a new section could start on line 309, corresponding to Fig. 4, and introducing the IFN experiments.  

-        - The cause of increased viperin expression in high glucose conditions is only touched briefly in the discussion section (line 367). Could this be elaborated somewhat? Did the authors evaluate the relation between ZIKV replication and expression of other IFGs upregulated in HGC such as TLR 4 or angiotensin II? (references 23-24) Could other factors involved in the interferon-viperin pathway play a role as well? Could possible limitations of the study be explained in the discussion?

-        - Some formatting mistakes have occurred in the figures, please check (Fig. 2D, 3A, 4A, 4C)

Specific remarks (line or figure numbers)

-        52: Here, although it is mentioned later in the manuscript, the Van der Hoek et al. (2017) reference is missing as the first to report the role of viperin in ZIKV infection.

-        72: β

-        186: could the number of included technical replicates be specified as well?

-        Fig 1B: could the flow cytometry results be presented  as scatterplot or histogram?

-        Fig. 1C: What is the explanation that less cells are observed in the 5.6 mM condition? Also, which Ab is used to stain nuclei (same for Fig. 2A)? Could this be specified in M&M or fig caption.

-        Fig. 1: so is it a correct interpretation that, based on these results, more cells express viperin in hgc conditions, rather than the amount of viperin per cell increases?  How do the authors explain this?

-        217: should be HK-2NGC

-        227: is this observation according to the literature? What is the explanation that untreated cells are more healthy than cells expressing a truncated but inactive viperin? On line 377, viperin is described as having ‘essential’ functions. This is paraphrased somewhat unclear, since cells without viperin are also healthy.

-        Fig. 2B: why are here the flow cytometry results quantified, and in Fig. 1B the immunofluorescence results? Could this be homogenized?

-        Fig. 2C: is it possible to include immunofluorescence images with ZIKV-GFP and stained viperin? This could add interesting information on the (co-)localization of viperin and ZIKV.

-        Fig 2D: This figure lacks a triple negative control (uninfected untransfected cells). Also, the authors do not give an explanation for the decreased tubulin under transfection conditions. Is this tubulin control used to normalize results?

-        Fig 2E (continuing on previous remark): compared to what are these results normalized? This should be stated in the text. Since tubulin is decreased, results should be compared to this for correct interpretation.

-        282: is this effect completely abolished when supernatant is added post-infection? (in congruency with the experiments of Fig 3D-E)

-        288: what is this observation suggesting, since it is defined as ‘interestingly’, could this be explained in the text.   

-        Fig. 3F: specify in the figure legend that conditions 3-6 are infected. Also, what is meant with the 24, 48, or 72h period? This is unclear from the text.

-        Fig. 4A-B: why is there an IFN increase in the mock or mut-transfected cells?

-        Fig. 4C: tick to be removed on x-axis

-        349: perhaps state in the figure caption that other differences were not significant.

-        402: could possible future experiments be introduced here?

Author Response

We thank the reviewer for valuable suggestions.

General remarks

  • - The structure of the manuscript is somewhat unclear. Could the section numbers and figures be homogenized? For example, Fig. 2 could be split in two separate figures according to section 3.2 and 3.3 (Fig.2 A-C and Fig. D-F), since this involves either NGC or HGC cells. Also, a new section could start on line 309, corresponding to Fig. 4, and introducing the IFN experiments. 

As recommended by reviewer 2, we divided figure 2 into 2 new figures (fig. 2 with transfected HK-2NGC and fig.3 with HK-2HGC) and we generated a new section 3.5 in the result section (in the new version of the manuscript line 367).

  • - The cause of increased viperin expression in high glucose conditions is only touched briefly in the discussion section (line 367). Could this be elaborated somewhat? Did the authors evaluate the relation between ZIKV replication and expression of other IFGs upregulated in HGC such as TLR 4 or angiotensin II? (references 23-24) Could other factors involved in the interferon-viperin pathway play a role as well? Could possible limitations of the study be explained in the discussion?

As stated in the introduction and discussion, there are links in the literature between high glucose environments and transcription factors (IRF1/3), which have also been involved, in other studies, with the upregulation of VP. Whether this sequence of activation also works in our model would be interesting to test but is further the scope of the current manuscript. We also tested for the upregulation of other ISG mRNAs (ISG15, 54 and 56) in our conditions but failed to detect any changes compared to controls. It seems that, somehow, VP increase could be, not really exclusive, but at least part of a “specific” upregulation. These observations are also very speculative and we don’t think it would help to add them in the manuscript. As we observed a nice correlation between VP upregulation and ZIKV inhibition upon high glucose concentration, we only focused our work on this very interesting finding. However, our data cannot exclude the contribution of other factors, although a major effect is observed with VP.

  • - Some formatting mistakes have occurred in the figures, please check (Fig. 2D, 3A, 4A, 4C)

Changes have been made to the figures.

Specific remarks (line or figure numbers)

52: Here, although it is mentioned later in the manuscript, the Van der Hoek et al. (2017) reference is missing as the first to report the role of viperin in ZIKV infection.

We added the reference in the introduction section line 52.

  • 72: β

Change has been made (line 73).

-        186: could the number of included technical replicates be specified as well?

    Change has been made (line 192).

  • Fig 1B: could the flow cytometry results be presented as scatterplot or histogram?

The flow cytometry results are represented as histogram with the SD on it.

  • 1C: What is the explanation that less cells are observed in the 5.6 mM condition? Also, which Ab is used to stain nuclei (same for Fig. 2A)? Could this be specified in M&M or fig caption.

We counted ten nuclei at 5.6 mM and 13 nuclei at 25mM. This was simply due to the field chosen in the figure (growth kinetics were done in our previous paper to avoid any bias). We took in consideration cell growth kinetics of HK-2 cells under each condition (as previously described by Reslan et al., 2020 / ref 23). A sentence explaining this matter was added in the method section (line 82). Nuclei were stained with DAPI (DAPI is a fluorescent stain that binds to DNA) (line 132).

  • 1: so is it a correct interpretation that, based on these results, more cells express viperin in hgc conditions, rather than the amount of viperin per cell increases?  How do the authors explain this?

In figure 1C, same signal intensity in NGC and HGC is observed by immunofluorescence. More precisely, the images in figure 1C are representative of 10 images taken in each condition. The percentage of VP positive cells presented in figure 1D is the result of the counting of VP positive cells in all 10 images. This correlates well with flow cytometry results (fig.1B) where the MFI is in line with the percentage of positive cells quantified in Fig.1D

  • 217: should be HK-2NGC

Change has been made line 233 (in the new version of the manuscript).

  • 227: is this observation according to the literature? What is the explanation that untreated cells are more healthy than cells expressing a truncated but inactive viperin? On line 377, viperin is described as having ‘essential’ functions. This is paraphrased somewhat unclear, since cells without viperin are also healthy.

We understand the concern of reviewer 2. The toxicity of the truncated form of viperin on cells is not yet clearly described (first observation made in Vanwalscappel et al., 2019 / ref 17). One possible hypothesis is that the truncated form, once over-expressed, dominates in the cell over the wild-type full-length protein and dysregulates cellular functions, even in the presence of a low quantity of this protein.

  • 2B: why are here the flow cytometry results quantified, and in Fig. 1B the immunofluorescence results? Could this be homogenized?

We understand the concern of reviewer 2. In figure 1B, the MFI (Mean Fluorescent Intensity) represents the quantification of cells by flow cytometry and not the IF result. The results presented in 1B and 2B were obtained by flow cytometry, and represented as MFI and % of positive cells, respectively.

  • 2C: is it possible to include immunofluorescence images with ZIKV-GFP and stained viperin? This could add interesting information on the (co-)localization of viperin and ZIKV.

We greatly appreciate the work of the reviewer to help us to improve the quality of our manuscript. We would like to mention that the GFP is not a fusion- or a trans-membrane protein so it is expressed and released into the cytoplasm. It is not the protein that interact with viperin but rather other viral proteins such as NS3 protein of the virus. This has been shown in the literature (ref 16 in the manuscript)

Fig 2D: This figure lacks a triple negative control (uninfected untransfected cells). Also, the authors do not give an explanation for the decreased tubulin under transfection conditions. Is this tubulin control used to normalize results?

In other experiments, not shown in this manuscript, we were not able to detect the Viperin protein in the negative control (uninfected-untransfected cells) in western blot analysis in absence of ZIKV infection even when adding the MG132 (Proteasome inhibitor). Western blotting experiment showed that only when cells were infected with ZIKV, the VP protein was detected. This is due to the short stability of the protein in the cells. Same observation was described by others (Yuan et al., 2020 doi : 10.1016/j.molcel.2019.11.003). Results were normalized to tubulin in the western blotting experiment.

  • Fig 2E (continuing on previous remark): compared to what are these results normalized? This should be stated in the text. Since tubulin is decreased, results should be compared to this for correct interpretation.

We agree with the reviewer 2 on this matter. Results were normalized to tubulin in the western blotting experiments. Thus, a sentence explaining this point was added in the legend of figure 3 and the result section (line 293).

  • 282: is this effect completely abolished when supernatant is added post-infection? (in congruency with the experiments of Fig 3D-E)

The answer is in the result section (Line 336 to 338): “ In contrast, when added 2 hours post infection, supernatants from rVPwt transfected HK-2 cells had only a moderate and not significant antiviral effect (now Figure 4D), and this ZIKV-inhibiting effect was completely lost 6 h after infection (now Figure 4E) ”.

  • 288: what is this observation suggesting, since it is defined as ‘interestingly’, could this be explained in the text.   

Our results are interesting because it's a time-dependent mechanism, the more the cells were incubated, the more they accumulate IFN in the supernatant, the more we see the paracrine antiviral effect. This is due to high stability of IFN.

  • 3F: specify in the figure legend that conditions 3-6 are infected. Also, what is meant with the 24, 48, or 72h period? This is unclear from the text.

We agree with reviewer 2 on this matter. To clarify this issue, the following sentence was added in the result section 3.4 (line 336): “ For this purpose, we transferred supernatants of HK-2HGC cells collected at 24h, 48h or 72h period, on naive HK-2NGC cells at different time points (2h, 6h and 24h) before ZIKVGFP infection. »

We already mentioned in the legend of figure 4F (in the new version of the manuscript) that in all conditions the cells were infected with ZIKVGFP

  • 4A-B: why is there an IFN increase in the mock or mut-transfected cells?

We already mentioned this in the discussion (line 476) : « we observed a moderate IFN-β production in mock and rVPmut transfected cells which has to be related to the lipofectamine effect (reported by Guo et al / 40) ».

  • 4C: tick to be removed on x-axis

Change has been made.

  • 349: perhaps state in the figure caption that other differences were not significant.

Change has been made (line 426).

  • 402: could possible future experiments be introduced here?

We greatly appreciate efforts of the reviewer to help us to improve the quality of the discussions. However, we think it will be too speculative to propose any additional experiments.

Reviewer 3 Report

The manuscript further investigated a previously described phenomenon that HK2 cells cultured in high-glucose medium were relatively resistant to Zika virus infection. The authors claimed that the data indicated that induction of viperin expression was responsible for the antiviral phenomenon. They also showed that the condition induced type I interferon response, and concluded that viperin probably induced the interferon response. Although the manuscript tried to investigate the mechanism involved in high-glucose medium-induced antiviral state, most of the experiments were on the effect of viperin and not on how high-glucose condition induced viperin. 

As both viperin and interferon were induced under the high-glucose condition, it is not possible to conclude which one was the cause of the other. In fact, it is more plausible to assume that viperin was induced by the interferon response, which was induced by the high-glucose condition. It is not surprising that knocking down viperin resulted in an increase in viral titer as viperin is an important anti-viral mechanism of interferon. It is hard to understand whether the rescue by si-viperin was complete as the effect of glucose itself was not shown in the experiment. The effect of high glucose on interferon response was previously reported in human PBMC (Hu et al, 2018). More importantly, the effect was reported to be biphasic and concentration-dependent. This brings  the next concern.

The next and a more important concern is whether this phenomenon is relevant to any in vivo situations or just a laboratory phenomenon without clinical implication. The high-glucose condition at 25 mM equivalent to 450 mg/dl is very high even for diabetic patients and the 10-day adaptation time requirement to see the effect makes it very unlikely to be relevant to in vivo conditions. To show that the phenomenon is relevant, lower concentrations of glucose should be tested. In addition, more cell lines especially those related to the viral pathogenesis should be tested. 

To conclude that the high-glucose condition directly induced viperin and not through interferon, a mechanistic link that exclude interferon effect needs to be shown. Otherwise, the glucose-induced antiviral state could be only concluded to be mediated by the innate interferon and viperin mechanisms with possible involvement of other interferon-induced antiviral mechanisms. 

Author Response

The manuscript further investigated a previously described phenomenon that HK2 cells cultured in high-glucose medium were relatively resistant to Zika virus infection. The authors claimed that the data indicated that induction of viperin expression was responsible for the antiviral phenomenon. They also showed that the condition induced type I interferon response, and concluded that viperin probably induced the interferon response. Although the manuscript tried to investigate the mechanism involved in high-glucose medium-induced antiviral state, most of the experiments were on the effect of viperin and not on how high-glucose condition induced viperin. 

As both viperin and interferon were induced under the high-glucose condition, it is not possible to conclude which one was the cause of the other. In fact, it is more plausible to assume that viperin was induced by the interferon response, which was induced by the high-glucose condition. It is not surprising that knocking down viperin resulted in an increase in viral titer as viperin is an important anti-viral mechanism of interferon. It is hard to understand whether the rescue by si-viperin was complete as the effect of glucose itself was not shown in the experiment. The effect of high glucose on interferon response was previously reported in human PBMC (Hu et al, 2018). More importantly, the effect was reported to be biphasic and concentration-dependent. This brings the next concern.

The next and a more important concern is whether this phenomenon is relevant to any in vivo situations or just a laboratory phenomenon without clinical implication. The high-glucose condition at 25 mM equivalent to 450 mg/dl is very high even for diabetic patients and the 10-day adaptation time requirement to see the effect makes it very unlikely to be relevant to in vivo conditions. To show that the phenomenon is relevant, lower concentrations of glucose should be tested. In addition, more cell lines especially those related to the viral pathogenesis should be tested. 

We thank the reviewer for valuable suggestions.

We agree with the reviewer that the 25mM glucose concentration is very high and is only found in uncontrolled diabetic patients leading to diabetic coma. However, in the present work, glucose concentration were chosen based on our previous study (Reslan A et al, IJMS 2021- PMID: 33801335) and on published works in which the 25mM glucose concentration is the most frequent used concentration to investigate hyperglycemia on HK-2 kidney cells and thus modelling possible implications in diabetic kidney disease (Prontelli et al, FASEB J 2017-PMID: 27881486; Hills CE et al, Am J Physiol Renal Physiol-PMID: 19091788; Hills et al, Diabetologia 2012-PMID: 22215279; Yaghobian D et al, Clin Exp Pharmacol Physiol 2016-PMID: 26414003; Ma Y et al, J Physiol Biochem 2020-PMID: 32424454).

Please, note that higher concentrations have been used i.e 30mM (Danesh FR et al, PNAS 2002-PMID 12048257; Peng F et al, Diabetes 2008-PMID 18356410), 60mM (Gu L et al, Chem Pharm Bull 2013-PMID: 23812394).

To clarify this point we have added the following sentence in the method section (line 82): “Glucose concentrations were chosen based on our recent study (Reslan A IJMS 2020 / ref23) and on previous published works (Hills et al, ref 26; Yaghobian D et al, ref 27; Prontelli et al, ref 28; Ma Y et al, ref 29).”

To conclude that the high-glucose condition directly induced viperin and not through interferon, a mechanistic link that exclude interferon effect needs to be shown. Otherwise, the glucose-induced antiviral state could be only concluded to be mediated by the innate interferon and viperin mechanisms with possible involvement of other interferon-induced antiviral mechanisms.

We greatly appreciate the efforts of reviewer 3 to help us to improve the quality of our manuscript. In our present study, we aimed to investigate the involvement of viperin in the antiviral effect observed under HGC to explain ZIKV infection inhibition in HK-2 kidney cells shown in our previous study (Alawiya et al., 2020 /ref 23). Indeed, we decided to focus our study on the link between VP and ZIKV infection inhibition, as we already had the interesting observation of the bystander effect.

Our experiments pushed us to study the antiviral paracrine effect of the supernatant of VP-overexpressing cells. -Interestingly, upon VP overexpression, cells were able to communicate through soluble connectors to enhance the antiviral effect. We could demonstrate from our overexpression experiments (HEK blue and ELISA) that recombinant VP is able to induce interferon production and that this IFN production takes a major role in the observed ZIKV infection inhibition.

In order to confirm that viperin was the inducer of the antiviral effect under HGC, we silenced VP gene and we observed a remarkable increase in the amount of ZIKV E protein by western blotting which was not observed in NGC. To clarify this interesting point raised by reviewer 3, we added the following sentence in the result section (line 295) : « which was not observed in NGC condition where the level of ZIKV E protein did not change before and after transfection with si-VP (data not shown) ».

Finally, in the literature, it is well described how viperin upregulation depends on functional IFN signaling pathway (ref 3 et ref 4). However, it has also been described other  pathways independent, more direct and faster than the IFN-dependent pathway (reviewed in Rivera-Serrano et al., 2020 Annual Review of Virology), via IRF1 (interferon regulatory factor 1), activator protein 1 (AP-1) and IRF3 (interferon regulatory factor 3) (ref 2, 5 et 6) . Interestingly, these transcription factors have been described to be overexpressed under HGC conditions in HK-2 cells.

Round 2

Reviewer 1 Report

The authors have responded to all questions emitted by this reviewer. 

Author Response

Thank you for your valuable review

Reviewer 3 Report

The authors admitted that there are limitations in their data and interpretation, but did not make any changes or additional works that were requested. 

Author Response

We recognize and understand the concern of reviewer 3 and to address his comments we added the following sentences in the conclusion part:

 line 472: Interestingly we could demonstrate from our overexpression experiments (HEK blue and ELISA) that recombinant VP, in HK-2NGC cells, was able to induce interferon production and that this IFN production takes a major role in the observed ZIKV infection inhibition.

 Line 479: However, in HK-2HGC cells, our data do not allow to clearly establish which comes first viperin or IFN-β. Indeed, we are facing a feedback loop since on one hand it has been shown that HGC can stimulate both IFN-β and viperin expression and on the other hand it is also known that viperin can stimulates IFN-β and vice versa. Although we demonstrated that si-viperin reduced the inhibitory effect on ZIKV replication we cannot ruled out the possibility of the intervention of IFN-β. So further studies blocking interferon are needed to clarify this point.
